# Cyberbullying and Social Anxiety: A Latent Class Analysis among Spanish Adolescents

**DOI:** 10.3390/ijerph17020406

**Published:** 2020-01-08

**Authors:** María C. Martínez-Monteagudo, Beatriz Delgado, Cándido J. Inglés, Raquel Escortell

**Affiliations:** 1Department of Developmental Psychology and Didactic, University of Alicante, 03690 Alicante, Spain; maricarmen.martinez@ua.es; 2Department of Health Psychology, Miguel Hernandez University of Elche, 03202 Alicante, Spain; cjingles@umh.es; 3Faculty of Education, International University of La Rioja, 26006 Logrono, Spain; raquel.escortell@unir.net

**Keywords:** cyberbullying, victimization, aggression, social anxiety, adolescence

## Abstract

Cyberbullying is a common social maladjustment that has negative repercussions on the wellbeing and development of adolescents, but numerous questions remain as to the relationship between cyberbullying and social anxiety in adolescence. This study analyzes cyberbullying profiles (screening of harassment among peers) and assesses whether these profiles vary with respect to the level of social anxiety (social anxiety scale for adolescents). The sample consisted of 1412 Spanish secondary education students aged 12 to 18 (*M* = 14.36, *SD* = 1.65). Latent class analysis and ANOVA were performed. Analyses revealed three profiles: high cyberbullying (high victimization, aggression, and aggression-victimization), low cyberbullying (moderate victimization, aggression, and aggression-victimization), and non-cyberbullying. The cyberbullying patterns varied significantly for all social anxiety subscales. Students with the high cyberbullying profile (bully–victims) presented high scores on social avoidance and distress in social situations in general with peers, whereas these students presented lower levels of fear of negative evaluation and distress and social avoidance in new situations as compared to the low cyberbullying (rarely victim/bully) and non-involved student profiles. Implications for psychologists and educational counselors and cyberbullying preventive interventions are discussed.

## 1. Introduction

Over the past decade, a major increase has been seen in news of bullying carried out by school-aged children using the new information and communication technologies (ICT). The ICT and social networks have become indispensable communication tools, especially for youth. This widespread use has offered many advantages; however, it has also led to some new violent behaviors resulting from the inappropriate use of these technologies. So, some students having a great domain of the ICT have taken advantage of these new virtual scenarios to engage in aggressive behavior towards their peers (such as insults, humiliation, coercion, the publication of confidential information, threats, denigration, violation of privacy, social exclusion, the spreading of rumors, identity theft, the dissemination of physical assaults, etc.). This phenomenon, known as cyberbullying, is defined as “a type of aggressive and intentional behavior that repeats frequently over time through the individual or group use of electronic devices with a victim that is unable to easily defend him/herself” ([1], p. 376).

The prevalence of cyberbullying has varied considerably in the studies that have been carried out until now. International reviews have reported mean prevalences ranging from 4% to 36% for cybervictimization and 16 to 18% for cyberaggression [2,3]. In a recent meta-analysis, Modecki, Minchin, Harbaugh, Guerra, and Runions (2014) found variations in prevalence ranging from 5 to 32% for cyberaggressors (mean of 16%) and between 2 and 56% for victims of cyberbullying (mean of 15%) [4]. These large variations may be due to differing conceptualizations of cyberbullying, the cut-off point criteria used to establish the frequency, the time framework established (an incident taking place during the past two months, last year, at any time, etc.), the type of methodology used, sample age range, etc.

### 1.1. Roles in Cyberbullying

The scientific literature has established three main roles with regard to this issue: victims, aggressors, and non-involved [5,6], with this being the most parsimonious classification. It is possible for aggressors to have previously been victims, with these students becoming aggressors in an attempt to earn a reputation of being strong and capable of defending themselves, and thus, the aggressor/victim role is created [6,7,8,9]. Studies have analyzed student roles in cyberbullying through the creation of cut-off scores that are based on statistical distributions that permit the assignment of the participants to one of these roles [5,10,11]. So, modification of the cut-off points alters the number of students belonging to a specific group, such that the stricter the established cut-off point, the lower the proportion of student aggressors [10], suggesting that these cut-off points may be relatively arbitrary.

This problem may be overcome using person-centered analytical approaches, such as cluster analysis and latent class analysis (LCA). With these analyses, student groups are generated based on specific indicators, permitting the creation of distinct groups based on the students’ real participation, with members of the same group having similar experiences that are distinct from those of other groups to which they do not belong. So, Aoyama, Bernard-Brak, and Talbert (2011), using cluster analysis with a sample of 133 US adolescents, identified four groups of roles involved in cyberbullying. The majority of the sample belonged to the “least involved” group (51.1%), 12.8% were “highly involved as bully and victim”, 10.5% were “more bully than victim”, and 9.8% were “more victim than bully” [12]. Along these lines, Schultze-Krumbholz et al. (2015), using LCA in an extensive sample of 6260 youth from six European countries, found that the majority of the sample belonged to the “non-involved” group (70.1%), while the “bully/victim” group was made up of 26.1% of the students and, a last group, the so-called “perpetrator with mild victimization” group, consisted of 4% of the selected sample [13].

Barboza (2015) used LCA to identify four categories: “highly victimized by both bullying and cyber bullying behaviors” (3.1%); “victims of relational bullying, verbal bullying, and cyber bullying” (11.6%); “victims of relational bullying, verbal bullying, and physical bullying but were not cyber bullied” (8%); and “non-victims” (77.3%) [14]. Hollá (2016), found three groups of students using LCA in a sample of 1619 Slovakian children and adolescents aged 11 to 18. Here, 52.9% of the students belonged to the “uninvolved” group while 42.7% were part of the “victims” group and 4.4% belonged to the “victims–aggressors” group [15].

In a more recent study, Betts, Gkimitzoudis, Spenser, and Baguley (2017), using a sample of 440 British students aged 16 to 19, identified four student profiles using a cluster analysis technique: “not involved” (33%), “rarely victim and bully” (40%), “typically victim” (26%), and “retaliator” (1%) [16]. In a subsequent study, Schultze-Krumbholz, Hess, Pfetsch, and Scheithauer (2018) used LCA on a sample of 849 German students (11 to 17 years of age), determining five groups: “prosocial defenders”, “communicating outsiders”, “aggressive defenders”, “bully–victims”, and “assistants” [17]. A summary of studies on cyberbullying with a person-centered analytical approach are presented in Table 1.

So, past empirical research supports the presence of distinct profiles in relation to cyberbullying. However, the results of these studies differ, most likely due to the distinct conceptualizations of cyberbullying, the type of methodology used, or the frequency considered necessary to consider the behavior “cyberbullying”. These inconclusive results support the need to continue analyzing the roles in cyberbullying to offer greater clarification about its functioning and to develop actions aimed at the effective prevention of the cyberbullying.

### 1.2. Cyberbullying Roles and Social Anxiey

Numerous studies have demonstrated the negative consequences of cyberbullying on all of the individuals involved [18,19]. So, although the most evident effects tend to be found on the victims, there is also an increased risk of suffering from psychosocial and emotional imbalances for aggressors and the non-involved individuals. The bidirectional relationship between cyberbullying and internalizing problems has been widely corroborated [20,21,22,23,24]. These problems include social anxiety, which involves a fear of negative evaluation, and general and specific social avoidance of new situations or individuals [25].

Various studies have strongly corroborated how victims of cyberbullying present high levels of social anxiety [26,27,28]. Navarro, Serna, Martínez, and Ruiz-Oliva (2015) suggest that students having high levels of social anxiety and, therefore, limited social skills in face-to-face interactions, may turn to the Internet and social networks to communicate and establish new friendships, interacting more frequently with strangers and, therefore, being more often exposed to on-line victimization [29]. On the other hand, distinct studies have indicated that on-line bullies tend to choose their victims from those individuals who have high levels of social anxiety, and therefore, who are less able to defend themselves [29,30,31]. Other authors have suggested that a bi-directional cycle is produced, in which students suffering from social anxiety are exposed to higher levels of victimization and repeated bullying may increase their already high levels of social anxiety [32].

Less attention has been paid to the assessment of social anxiety in those students who are cyberbullies. Kowalsky et al. (2008) found that cyberbullies reported similar levels of social anxiety as the non-involved students [28]. Similarly, Pabian and Vandernbosch (2016) concluded that social anxiety is not a risk factor for engaging in cyberbullying behavior [33]. Other studies, however, have found high levels of general anxiety in cyberbullies [34]. Harman, Hansen, Cochran, and Lindsey (2005) suggested a possible relationship between social anxiety and being a perpetrator of cyberbullying, indicating that those adolescents with limited social skills in face-to-face interactions may use the digital media to escape from their social fears, feeling less restricted and less inhibited, and therefore may be more likely to engage in aggressive on-line behavior [35]. But currently, few studies exist on social anxiety in student cyberbullies.

As for the aggressor/victim role and its relationship with social anxiety, Kowalski et al. (2008) found that victims of cyberbullying have high levels of social anxiety, followed closely by the aggressor–victim group. Similarly, they found that this last group of students revealed higher levels of general anxiety than the “pure” cybervictim students and students who were not involved [28]. Along the same lines, Kowalski and Limber (2013) found that the group of aggressors–victims reported high levels of anxiety [36]. Recently, Fahy et al. (2016) performed a longitudinal study on a sample of 2480 adolescents, aged 12 to 13, finding that the cybervictim and cyberbully–victim groups had a higher probability of reporting social anxiety one year after experiencing cyberbullying [21].

Therefore, the evidence underscores that adolescents involved in cases of cyberbullying (victims, aggressors, or bully–victims) manifest higher levels of social anxiety. Besides, low social skills and interpersonal anxiety can promote the use of ICT as means of social contact and increase the risk of being victimized or perpetrating electronic harassment. However, it remains to be clarified whether the different groups/roles differ in the symptoms of social anxiety (as fear of negative evaluation, distress, social avoidance, etc.) and in different social situations.

### 1.3. The Present Study

Based on all of this, our study has two objectives. First, using a sample of Spanish adolescents, it aims to identify whether or not there are combinations of different cyberbullying roles that lead to distinct profiles, which are defined based on a greater or lesser weight of each of the cyberbullying dimensions (victimization, aggression, and aggression-victimization) within each profile. Second, examine the differences in social anxiety using distinct cyberbullying profiles. The following hypotheses were created based on past empirical research carried out in the adolescent population:

**Hypothesis** **1(H1).**
*The following cyberbullying profiles are expected to be found: (1) victims (high scores on victimization and low scores on aggression and aggression-victimization); (2) bullies (high scores on aggression and low scores on victimization and aggression-victimization); and (3) bully–victims (high scores on aggression-victimization and low scores on victimization and aggression); (4) not involved (low scores on aggression, victimization, and aggression-victimization).*


**Hypothesis** **2(H2).**
*It is anticipated that the group with high scores on victimization and low scores on aggression and the group with mainly high scores on aggression and victimization will have more social anxiety than the other groups.*


## 2. Materials and Methods

### 2.1. Participants

The sample consisted of secondary education students from the autonomous community of Valencia (Spain). Two-stage random sampling was conducted. In the first stage, 18 public and charter/semi-private secondary schools were randomly selected. Once the schools were selected, in the second stage of sampling, four classes were randomly selected from each school. Due to the random sampling method, the socioeconomic status and ethnic composition of the overall sample were assumed to be representative of the community population. Initially, 1462 students were recruited, of which 50 (3.4%) were excluded due to failure to provide parental consent or errors found in their responses. The final sample consisted of 1412 Spanish secondary education students (653 males and 759 females), aged 12 to 18 (*M* = 14.36; *SD* = 1.65). Student distribution based on gender and academic year of study was as follows: 244 in 7th grade (118 males and 126 females), 253 in 8th grade (107 males and 146 females), 250 in 9th grade (123 males and 127 females), 242 in 10th grade (E.S.O.) (112 males and 130 females), 278 in 11th grade (127 males and 151 females), and 145 in 12th grade (66 males and 79 females). Using the Chi-squared test of homogeneity of the frequency distribution, it was verified that there were no statistically significant differences between the groups gender x class year (*x*^2^ = 7.21; *p* = 0.29).

### 2.2. Instruments

Screening of harassment among peers (SPH, [37]) is a self-reporting instrument that permits assessment of bullying and cyberbullying behavior. In this study, only the scores for victimization, aggression, and aggression-victimization were used from the cyberbullying scale. This scale assesses 15 bullying behaviors carried out via electronic means (sending offensive or insulting messages, making offensive calls, spreading photos or videos over YouTube, making anonymous calls to scare, threaten, or bribe), allowing for the identification of victims, aggressors, aggressors-victimized, and observers of cyberbullying. The cyberbullying questionnaire consists of a total of 45 items and a Likert-like response format with four options (1 = never; 4 = always). The response system is triangular, since the assessed individual should identify whether or not he/she has suffered from the 15 bullying behaviors as a victim, if he/she has engaged in these behaviors as a perpetrator, or if he/she has witnessed them being carried out on another individual or have been aware of their occurrence during the past year. The psychometric studies carried out in the original study support the test’s internal consistency (*α* > 0.82) [37]. In this study, the internal consistency rates of the subscales were found to be adequate: victimization (*α* = 0.95), aggression (*α* = 0.96), and aggression-victimization (*α* = 0.98).

Social anxiety scale for adolescents (SAS-A, [25]) is a self-reporting questionnaire that measures social avoidance, fears, and worries of adolescents during social situations. It consists of 22 items (18 of which are self-descriptions and 4 are neutral items that are not considered in the scoring) which are responded to using a Likert scale of 5 points (1 = never; 5 = always). The SAS-A consists of three subscales: fear of negative evaluation (FNE; 8 items) which measures the fears and concerns due to potentially negative peer assessments (e.g., “I’m afraid that others will not like me”); social avoidance and distress in new situations (SAD-New; 6 items), which assesses social avoidance and discomfort felt during new social situations and with non-familiar individuals (e.g., “I feel nervous when I’m around certain people”); and social avoidance and distress-general (SAD-General; 4 items), which assesses social inhibition, discomfort, and distress that, in general, are experienced in social situations (e.g., “It’s hard for me to ask others to do things with me”). The questionnaire’s appropriate psychometric properties have been confirmed in samples of American, Spanish, Chinese, Portuguese, Finnish, and French adolescents, with internal consistency indicators (Cronbach’s alpha) exceeding 0.70 for the SAS-A subscales [38]. In this study, the values indicate an adequate reliability of the three subscales: FNE (*α* = 0.79); SAD-New (*α* = 0.75); and SAD-General (*α* = 0.70).

### 2.3. Procedure

Initially, meetings were held with the principals of the participating schools, explaining the study objectives and requesting their collaboration. Then, permission was granted from local and regional governmental institutions as well as the consent of the Ethics Committee of the university supporting the project (UA-2018-02-21), so that the study could be carried out. At this point, a letter was sent to the parents of the students requesting their written consent for their children’s participation in the study. The questionnaire was administered to the students in an anonymous and collective manner. Mean administration times were 15 min (SPH) and 15 min (SAS-A). Standards regarding research on human subjects were respected, in accordance with the ethical principles of the Declaration of Helsinki.

### 2.4. Statistical Analysis

Latent class analysis (LCA) was used to identify the distinct cyberbullying profiles. These profiles were established based on the aggregate scores of the distinct behaviors of victimization, aggression, and aggression-victimization of cyberbullying (Authors have also conducted LCA with all separately items of SPH and found similar latent classes as the LCA with three subscales). Because all the roles did not have the same number of items, the aggregate scores were transformed into z scores to calculate the LCA. Based on the profile presented by the students, they were included in one of these classes. The election of the number of classes needed to identify a better representation of the data was carried out using the lowest indicator of the Bayesian information criteria (BIC) and the Akaike information criterion (AIC) and the value closest to one for entropy [39] as the adjustment indices. Then, ANOVAs were performed to verify whether or not differences existed in social anxiety between the distinct groups and the post hoc Bonferroni test was used to determine which groups presented statistically significant differences. Finally, Cohen’s d (standardized difference between means) [40] was used to assess the magnitude of said differences. Its interpretation is as follows: 0.20 ≤ *d* ≤ 0.50, suggests a small effect size, 0.51 ≤ *d* ≤ 0.79 is moderate, and *d* ≥ 0.80 is a large effect size. The XLSTAT version 2019 and SPSS Statistics 26 programs were used for conduct LCA and ANOVAs, respectively.

## 3. Results

### 3.1. Cyberbullying Profiles

LCA was used, taking into account the scores of the three cyberbullying behaviors (victimization, aggression, and aggression-victimization). As seen in Table 2, the class obtaining the best fit for the BIC, AIC, and entropy indicators was that consisting of three profiles. The first profile, non-cyberbullying, consisted of a total of 603 students (42.70%) having very low scores on the subscales of victimization, aggression, and aggression-victimization, identified as “not involved”. The second profile, high cyberbullying, with 424 students (30.02%) having high levels of victimization, aggression, and aggression-victimization, identified as “bully–victims”. The third profile, low cyberbullying, consisting of 385 students (27.26%), had moderately low scores on the three analyzed subscales of cyberbullying, identified as “rarely victim and bully”. Figure 1 shows the LCA solution including the z scores for victimization, aggression, and aggression-victimization.

### 3.2. Inter-Group Differences in Social Anxiety

Table 3 shows the results of the ANOVAs comparing each trait of social anxiety between the three cyberbullying profiles. The analyses reveal statistically significant differences for the three SAS-A subscales. Specifically, students included in the low cyberbullying (“rarely victim and bully”) and the non-cyberbullying (“not involved”) profiles had significantly higher scores on fear of negative evaluation and social avoidance and discomfort in new social situations than students in the high cyberbullying (“bully–victim”) profile. On the other hand, adolescents in the profile with high scores (“bully–victim”) presented significantly higher scores on social avoidance and discomfort in social situations in general, as compared to students in the other two profiles. Furthermore, those in the low cyberbullying (“rarely victim and bully”) profile had significantly higher scores in discomfort and social avoidance in new situations and in social situations in general, as compared to the group of students who were not involved in cyberbullying.

The effect size for the differences found in the comparisons between the high cyberbullying (“bully–victim”) profile and the other profiles was high for the fear or negative evaluation subscale (*d* > 0.84), moderate for the discomfort and social avoidance in new situations subscale (*d* > 0.53), and low for the subscale of social avoidance in general (*d* < 0.48). The magnitude of the differences found between the profiles of low cyberbullying (“rarely victim and bully”) and non-cyberbullying was very small in all cases (Table 4).

## 4. Discussion

The first objective of this study was to analyze the cyberbullying profiles in adolescents using SPH scores for victimization, aggression, and aggression-victimization [37] using latent class analysis. The second objective was to examine the differences in social anxiety (fear of negative evaluation, anxiety, and social avoidance in new situations and discomfort and social avoidance in social situations in general) using distinct cyberbullying profiles.

In contrast to that established in the first hypothesis, in which it was expected to find four groupings based on the traditional concept of bullying (victims, bullies, bully–victims, and not involved), the results of the LCA suggest a better fit for the solution of the model with three classes: non cyberbullying (not involved), high cyberbullying, and low cyberbullying. Thus, a profile of adolescents who are not involved in cyberbullying cases is found (42.7%), scoring very low on victimization, aggression, and aggression-victimization; another group with high scores on behaviors corresponding simultaneously to cybervictims and cyberaggressors, or “bully–victims” (30%), and a third group with moderately low scores on cyberbullying behaviors, or the “rarely victims and bullies” (27.26%). These findings are in line with those found by Hollá (2016) and Schultze-Krumbholz et al. (2015) who used the LCA methodology to find three independent groups of cyberbullying [13,15]. However, they differ with respect to the composition and percentage of subjects included. The prevalence of “not involved” subjects obtained in this study is much lower than findings from past studies [13,14,15,17,20] which found rates of 52% to 77% of adolescents who were not involved in the cyberbullying cases. However, the percentage found by Betts et al. (2017), in a sample of youth aged 16 to 19 (33%) is higher [16]. Another discrepancy is found for the “bully–victim” profile, with the percentage of victims and bullies varying substantially from one study to another. So, we find studies that have a similar prevalence to that of our study [13] while in others [12,14,15,17] the prevalence of this profile is much lower (3–12%). The third profile found in this study, the “rarely victim and bully” group, is characterized by being involved to a greater degree in all of the cyberbullying behaviors (victimization, aggression, and aggression-victimization) as compared to the not-involved group, but not at the levels that are typical of the “bully–victim” group (high cyberbullying). For this profile, similarities and differences were found with respect to past study results. So, despite the fact that certain studies using Person-centered analytical approaches have failed to detect this profile [12,13,14,15,17], others have identified this as a group consisting of a high percentage (40%) of adolescents [16]. So, in comparison to the findings from past studies, and reinforcing the conclusions reached by Betts et al. (2017) the results of this study suggest that the frequency of participation in cyberbullying is perhaps the factor that best determines the student grouping for on-line bullying. This finding may be based on the very nature of cyberbullying, since exposure to cyberbullying is not limited to a specific space and time (e.g., school) as occurs with traditional bullying, but rather, depends on the time spent “connected” and the extent of adolescent participation in the cyberbullying situations.

As for the lack of an exclusive victim or aggressor profile in this study, despite differing from certain studies [15,17], this evidence appears to be based on the ever-increasing simultaneous nature of cyberbullying perpetration and victimization behaviors. As suggested by Schultze-Krumbholz et al. (2015), the perpetrators of cyberbullying are not free from suffering from ICT victimization experiences, since the victims may confront their cyberbullies more easily than in a physical environment, where there may be greater differences in status and power position between the individuals involved [13]. Furthermore, the lack of a “pure” aggressors or victims group and the presence of motivation for revenge may support the growing evidence, suggesting that youth can use ICT to intimidate others in response to the very cybervictimization that they have suffered, turning them into a so-called “bully–victim” [13,16].

To analyze the second study objective, three latent classes were considered, along with their mean scores on social anxiety. Inter-class differences were found on scores for fear of negative evaluation, discomfort, and social avoidance in new situations, and discomfort and social avoidance in social situations in general. The “bully–victim” profile (high cyberbullying) presented scores that were significantly higher in discomfort and avoidance in social situations in general, as compared to the “not involved” and “rarely victim and bully” profiles. These results partially confirm the second hypothesis that suggests that the group with high scores on aggression and victimization had greater social anxiety than the other groups, and reinforces findings from other studies that suggested greater levels of social anxiety in victims [26,27,28,33] and in bully–victims [28,36]. This suggests that adolescents who are more intensely involved in cyberbullying behaviors and who are aggressors/victims are more distressed, socially uncomfortable, and more likely to avoid most social situations in which they are expected to relate with peers. These typical behaviors of generalized social anxiety are more disabling over the long term [41] and may potentially cause social interaction problems, inhibiting social behavior with peers in real life environments and causing the individuals to take refuge in virtual scenarios (e.g., social networks, chats, on-line games) in an attempt to develop social relationships with others (who are often strangers). These dynamics of social interaction may affect the social development of the youth, often times damaging their learning of adaptive social skills and abilities, increasing the probability of their continuing to be victimized on-line and, therefore, maintaining the on-line bullying [29].

However, the adolescents in the “rarely victim and bully” and “not involved” profiles revealed higher levels of fear of negative evaluation and greater social avoidance in new social situations as compared to those in the “bully–victim” (high cyberbullying) profile. These unexpected findings are surprising, since, despite the fact that it was expected that the group with the highest scores of victimization and aggression (“bully–victim”) would score higher in social anxiety than the other groups, the results suggest that those in the “rarely victim and bully” profile are the most likely to avoid relations with peers who are strangers and to feel the most fear of being negatively evaluated (this characteristic is shared with the not involved profile). On the one hand, these results suggest that adolescents having a moderately low participation in cases of cyberbullying may present fear and concern that their peers will judge them negatively, leading them to avoid situations in which they must interact with others who they do not know well. This result may be explained by the impact of anxiety on minors during early stages of cyberbullying. So, it has been found that during early adolescence, anxiety in the case of social evaluation was the variable that was the most likely to explain being a victim, while anxiety in the face of school punishment was the factor that most likely determined being an aggressor or aggressor-victimized [42]. This reinforces evidence from past studies regarding the bidirectional and positive association between the symptomology of social anxiety and victimization [43,44,45] and the perpetration of cyberbullying [21,34,35,43]. So, adolescents who are less socially skilled in face-to-face interactions may use the digital media to hide from their social fears and feel less inhibited and restricted, potentially increasing their connectivity and participation in the Internet and the social networks, and thereby increasing their probability of engaging in or suffering from aggressive on-line behavior [46]. Thus, adolescents who are involved in moderately low levels of cyberbullying may have already demonstrated a low level of self-confidence and feelings of insecurity, characteristic of social anxiety, leading them to avoid social relationships with their peers.

On the other hand, the results suggest that adolescents who are not involved in cyberbullying, although not actively participating in these behaviors, may observe this on-line bullying of others, leading them to be concerned or distressed about the possibility of being criticized or made fun of themselves. Along these lines, some studies have identified problems of psycho-emotional adjustment in the cyberbullying observers [18,24,47] as well as a low self-concept and feelings of sadness, impotence, guilt, and fear. So, these negative emotional consequences may lead to a greater aversion and fear related to others who they do not know, for fear of becoming a victim [46].

This study has certain limitations that should be considered when carrying out future studies. First, the sample type used is a limitation, given that the results cannot be generalized to students from other education levels (primary or higher education levels) due to the different developmental characteristics of each level. Future studies should analyze whether or not the findings obtained differ (or not) at other academic levels. Second, given that this is a cross-sectional study, it is impossible to establish causal relationships; so, in the future, longitudinal studies with an experimental design should be carried out. Third, the “not involved” group is quite heterogeneous and tends to be made up of distinct behavioral and attitudinal profiles of “bystanders” [17]. The measurement and control of this issue should be the subject of further study. Finally, limitations also result from the lack of consensus regarding the definition of cyberbullying (e.g., roles and behaviors), and therefore, this definition should be updated in response to the constant technological advances being made in this area.

## 5. Conclusions

This study provides novel information that is of great relevance to the study of cyberbullying during the period of adolescence. On the one hand, it uses a person-centered analytical approach, LCA, surpassing the potential limitations of arbitrariness of the distinct cut-off points established for each measurement instrument. Also, it provides valuable information as to the grouping of adolescents into three cyberbullying profiles based on the frequency of their participation in behaviors of victimization, aggression and aggression-victimization, and not in the roles that have traditionally been identified in bullying. The findings from this study highlight the need to consider: a “not involved” profile, consisting of 42.7% of the adolescents who attain very low scores on the three cyberbullying behaviors; a profile with high scores on the three behaviors, the “bully–victims” who are one third of the adolescents and who present the most disabling of social anxiety symptoms over the long term, since they experience discomfort and avoid relationships in most social situations; and a third profile consisting of a high percentage of adolescents who are involved in cyberbullying, but in a minor way (27.3%), and that demonstrate feelings of fear of negative evaluation and avoidance of social situations with unknown individuals. These findings should be considered in terms of prevention and intervention in the psycho-educational area, since adolescents, even when only involved in these cyberbullying cases in a limited manner, may have emotional repercussions such as social fears and the avoidance of peer relations. These results lead to the possibility of improved characterization of classes or profiles for cyberbullying during adolescence, permitting the design of more effective preventive strategies. So, intervention programs that are intended to reduce the risk of cyberbullying in secondary education should consider the decrease in social anxiety and the learning of social skills and competencies for all identified profiles, even though those adolescents in the “bully–victim” profile may require a prioritized intervention to decrease their levels of discomfort and social avoidance in the scholastic environment.

## Figures and Tables

**Figure 1 ijerph-17-00406-f001:**
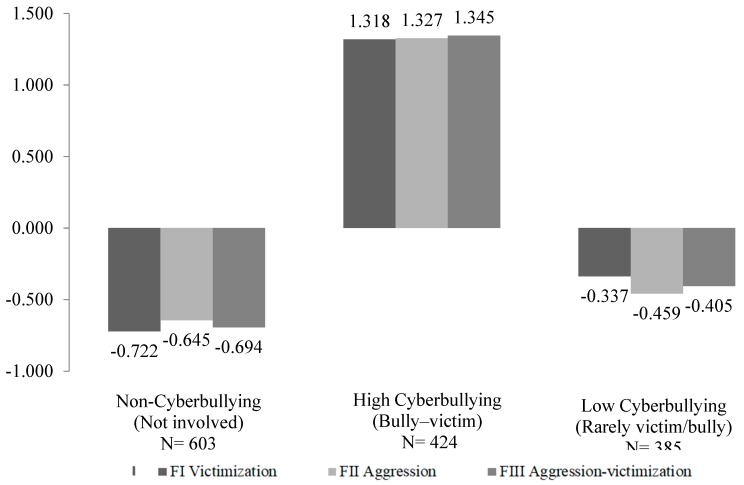
Graphic representation of the LCA solution.

**Table 1 ijerph-17-00406-t001:** Summary of the studies reviewed.

Source	Country	Subjects	Method	Classes
Aoyama et al., 2011	USA	133 high school students (*M_age_* = 15.7)	Cluster analysis	51.1% least involved 12.8% highly bully and victim 10.5% more bully than victim 9.8% more victim than bully
Barboza, 2015	USA	5589 students (aged 12–18)	LCA	77.3% non-victims 11.6% victims of relational and verbal bullying and cyberbullying 8% victims of relational, verbal and physical bullying 3.1% highly victimized by both bullying and cyberbullying
Betts et al., 2017	United Kingdom	440 students (aged 16–19)	Cluster analysis	33% not involved 40% rarely victim and bully 26% typically victim 1% retaliator
Hollá, 2016	Slovakia	1619 students (aged 11–18)	LCA	52.9% uninvolved 42.7% victims 4.4% victims–aggressors
Schultze-Krumbholz et al., 2015	Poland, Spain, Italy, United Kingdom, Germany, Greece	6260 students (aged 11–23)	LCA	70.1% non-involved 26.1% bully/victim 4% perpetrator with mild victimization
Schultze-Krumbholz et al., 2018	Germany	849 students (aged 11–17)	LCA	52.2% prosocial defenders 28.4% communicating outsiders 9.5% aggressive defenders 7.1% bully–victims 2.8% assistants

**Table 2 ijerph-17-00406-t002:** Fit indices of the latent class analysis (LCA).

No. of Classes	BIC	AIC	Entropy	Number of Parameters
2	2616.85	2548.56	0.973	13
**3**	**197.16**	**302.22**	**0.967**	**20**
4	1780.81	1922.63	0.964	27
5	2496.42	2675.03	0.948	34
6	3070.73	3286.13	0.952	41

BIC: Bayesian Information Criterion; AIC: Akaike Information Criterion; Values in bold revealing the best model fit.

**Table 3 ijerph-17-00406-t003:** Means and standard deviations of social anxiety between classes and statistical significance.

	Non-Cyberbullying		High Cyberbullying		Low Cyberbullying		*F*	*p*	η^2^
	*M*	*SD*	*M*	*SD*	*M*	*SD*			
FNE	20.43	4.19	16.83	4.39	20.94	4.40	117.06	0.00	0.142
SAD-New	16.28	4.42	14.05	3.92	16.93	4.14	54.88	0.00	0.072
SAD-General	8.24	3.32	9.71	2.53	8.89	3.35	27.53	0.00	0.038

FNE = Fear of negative evaluation; SAD-New = Social avoidance and distress with peers in new situations or with unfamiliar peers; SAD-General = Social avoidance and distress that was generally experienced in the company of peers.

**Table 4 ijerph-17-00406-t004:** Cohen’s d index to post hoc Bonferroni contrast between the means scores and the three classes in the factors of social anxiety.

	High Cyberbullying- Non-Cyberbullying	High Cyberbullying- Low Cyberbullying	Low Cyberbullying- Non-Cyberbullying
FNE	0.84 ***	0.94 ***	0.12
SAD-New	0.53 ***	0.72 ***	0.15 *
SAD-General	0.48 ***	0.28 **	0.19 **

* *p* < 0.05; ** *p* < 0.01; *** *p* < 0.001; FNE = Fear of negative evaluation; SAD-New = Social avoidance and distress with peers to new situations or unfamiliar peers; SAD-General = Social avoidance and distress that was generally experienced in the company of peers.

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
