# Peer review of "Cyberbullying and Social Anxiety: A Latent Class Analysis among Spanish Adolescents"

_ijerph, 2020, doi:10.3390/ijerph17020406_

Round 1

Reviewer 1 Report

This manuscript examines cyberbullying profiles and the association of the profiles to social anxiety levels. Using data collected from Spanish Secondary Education students, the authors identified three types of cyberbullying profiles. The strength of the manuscript is the detail discussion of the literature, methodology and discussion section. Although the study has some limitations (such as generalization issues), the authors acknowledge this limitation and provide recommendations for future research.

Reviewer 2 Report

Lines 34 - 36 provide examples of cyberbullying. Within the examples, the term "bullying" is listed as an example.  This seems redundant, as it is a given that cyberbullying is a form of bullying. 

In my opinion, the bulk of the introduction reads as a list of statistics that could easily be summarized visually.  There is a great deal of "noise" that muddies the authors' main points.

The categories in hypothesis 1 should be clarified.  Again, the writing style disrupts the flow of the information.

"Institutions" is misspelled in line 190.

The participants were drawn from two distinct types of institutional settings. In terms of the charter schools, it is important to determine the type of student served within. That is, there are charter schools for those who are sent to attend due to disciplinary issues within their previous public or private school setting.  For example, a charter school near my place of employment houses pregnant adolescent females, while another charter school was designed to assist aggressive teens.  As such, it is tough to apply the current findings to the general population. Were the authors to compare the two types of school, public and charter, and describe the differences in results, the research would be more credible. 

Reviewer 3 Report

Cyberbullying and social anxiety: A latent class 3 analysis among Spanish adolescents

Overall, I found this manuscript to be well-written. It is studying on the relations between cyberbullying and social anxiety, potentially shedding light on the outcomes of cyberbullying perpetration and victimization.  The literature review is concise and well-written.  I found that the major contribution of this study is the use of Latent class analysis (LCA) to identity underlying patterns.  However, as I stated below, there are major problems associated with the LCA the authors reported.  The problems may be resolved if the authors provide more details on the analytical procedures of LCA.

“there is also an increased risk of suffering from psychosocial and emotional imbalances for aggressors” – the authors suggested that aggressors may result in psychosocial and emotional problems. However, it is possible that these psychosocial/emotional problems caused aggressions (there are a lot of literatures on this relations).  When the authors made the argument that aggressions caused/resulted in psychosocial/emotional problems, did they have any experimental findings to support their claim? If not, the authors should recognize the bidirectional relations between these two variables.  This is also a problem in the discussion.  In interpreting the findings, the authors should highlight the possibility of both directions.  The authors did talk about the limitation, “Second, given that this is a cross-sectional study, it is impossible to establish causal relationships; so, in the future, longitudinal studies should be carried out to provide information over time” but they mistook longitudinal studies as causal study.  Longitudinal studies, if not designed experimentally, are still CORRELATIONAL study! The authors should not confuse the two. The authors mentioned that the schools were randomly sampled. Could the authors please provide details on how schools were randomly selected? Did the authors use certain algorithm to randomly select the schools regardless of the size and characteristics of schools? Also, 18 schools responded to participate, I assume? How many schools did the authors originally reach out to? Did all schools the authors randomly selected agreed to participate? “In this study, only the scores for Victimization, Aggression and Aggression-victimization were used from the cyberbullying scale.” – why did the authors only choose these three subscales? What were the subscales the authors did not use and how did the authors make the decision to exclude those subscales? . Please state what statistical tool and what version was used to conduct the analyses. More explanations and descriptions are needed for how the authors handled the cyberbullying items before putting them in the LCA model. In Figure 1, it seems that the authors used three means (one for victimization, one for aggression and one for victimization-aggression) to compute the classes.  Does this mean that the authors average the scores for each subscales and then put them in the LCA? If my understanding is correct, then I think the authors should re-do the LCA with all 45 items, instead of aggregated scores.  Also, the items were rated on 4-point Likert scales.  If the authors used LCA, did they keep the 4-point scale? These are all the details lacking in this manuscript. What the LCA results shows now is that high cyberbullying aggressors are also high cyberbullying victims, and they are much more socially anxious than the other low cyberbullying and no cyberbullying groups. The authors discussed this finding in the discussion, which I appreciated.  However, I think this is an important finding and it is far beyond our current understanding (that there are bullies and there are victims, and bullies are always bullies and victims are always victims).  Therefore, the authors’ LCA procedures and results should be clearly presented.  The authors should consider constructing a table showing how each of the 45 items falls into the LCA classes (again, I don’t think the authors should do aggregated scores in LCA – they should do the LCA with all 45 items!).  That is, shows the probability of how each item with a 4-point scale falls into the three classes (i.e. the output of your LCA tables, rather than a summary of class membership percentages that the authors currently reported). Also, in figure 1, y-axis was not labeled nor explained in the texts/figure captions, making the results of the LCA looks even more confusing. Before the authors conducted ANOVA, did they test if the groups (classes from LCA) meet the assumption of ANOVA, that is, did the groups have equal variance? I am concerned because the sizes of the groups were not quite equal. Table 3 can be improved by stating all effect sizes (instead of indicating n.s. for the non-significant one), and then mark those that are significant with asterisks (e.g., *p<.05, **p<.01, ***p<.001). This is because it is important to report effect sizes even for non-significant relations. Finally, a relatively minor problem lies in the scales of cyberbullying. The authors stated that “In this study, the internal consistency rates of the subscales were found to be adequate: Victimization ( = .95), Aggression ( = .96) and Aggression-victimization ( = .98).” – the alpha is extremely high, suggesting that it is unnecessary to use all 15 items per subscale to test their behaviors.  The authors may consider reducing the number of items by excluding items that are the same (very close to 1.0 correlation coefficients) and then use all the remaining items in LCA (that is, instead of using all 45 items, the authors may use less items, making the LCA easier to interpret).  

Round 2

Reviewer 2 Report

I commend you for thoroughly reviewing and addressing the recommendations.  However, I continue to stand by the recommendation that the studies reviewed in the introduction should be summarized in table form.  It's acceptable to keep the information as it is providing a visual (table) representation is added. 

Though this presents a different data set, this is the type of table to which I am referring:

Author Response

Thank you for your suggestion. We have not been able to visualize the table that you suggest us. So, we have made a summarize table that include the main results of the studies reviewed (Table 1). Therefore, all tables in the manuscript have been renumered. 

Reviewer 3 Report

I found the authors' revisions to be very satisfactory. The only thing I will add is that the authors may want to consider including their new LCA analysis reported in the "response to the reviewer" as a supplementary file, or at least as a footnote in the manuscript.  They may state that they have also conducted LCA with all 45 items and found similar latent classes as their LCA with 3 subscales. 

Author Response

Thank you for your suggestion. We have included a footnote in the manuscript (line 236) informing that "Authors have also conducted LCA with all separately items of SPH and found similar latent classes as the LCA with three subscales".
